# A One-Month Advanced Glycation End Products—Restricted Diet Improves CML, RAGE, Metabolic and Inflammatory Profile in Patients with End-Stage Renal Disease Undergoing Haemodialysis

**DOI:** 10.3390/ijms25168893

**Published:** 2024-08-15

**Authors:** Adamantia Aroni, Paraskevi Detopoulou, Demetrios Presvelos, Eirini Kostopoulou, Anastasios Ioannidis, George I. Panoutsopoulos, Sofia Zyga, Georgios Kosmidis, Bessie E. Spiliotis, Andrea Paola Rojas Gil

**Affiliations:** 1Laboratory of Basic Health Sciences, Department of Nursing, Faculty of Health Sciences, University of Peloponnese, 22100 Tripoli, Greece; adamantia.aroni@gmail.com (A.A.); tasobi@uop.gr (A.I.); georgios.kosmidis@goc.com.cy (G.K.); 2Haemodialysis Unit, General Hospital of Molaoi, 23052 Molaoi, Greece; dpresvelos@gmail.com; 3Department of Nutritional Science and Dietetics, Faculty of Health Sciences, University of Peloponnese, New Building, Antikalamos, 24100 Kalamata, Greece; viviandeto@gmail.com (P.D.); gpanouts@uop.gr (G.I.P.); 4Department of Clinical Nutrition, General Hospital Korgialenio Benakio, Athanassaki 2, 11526 Athens, Greece; 5Department of Paediatrics, Research Laboratory of the Division of Pediatric Endocrinology and Diabetes, University of Patras School of Medicine, 26504 Patras, Greece; eirini.kost@gmail.com (E.K.); besspil@gmail.com (B.E.S.); 6Laboratory of Nursing Research and Care, School of Health Sciences Department of Nursing, University of Peloponnese, 22100 Tripoli, Greece; zygas@uop.gr

**Keywords:** AGEs, chronic kidney disease, haemodialysis, diet

## Abstract

Exogenous and endogenous advanced glycation end products (AGEs) contribute to the pathogenesis and progression of renal disease. This is a one-month controlled dietary counseling trial that restricts nutritional AGEs in patients with end-stage renal disease (ESRD) undergoing haemodialysis (*n* = 22 participants in the intervention and *n* = 20 participants in the control group). Haematological, biochemical markers, the soluble form of the receptor for AGEs (sRAGE), and carboxymethyl lysine (CML) were measured at baseline and at follow-up. Mononuclear cells were isolated and the protein expression of RAGE and the inflammatory marker COX-2 was measured using Western immunoblotting. The intervention group presented a lower increase in CML compared to the control group (12.39% median change in the intervention vs. 69.34% in the control group, *p* = 0.013), while RAGE (% mean change −56.54 in the intervention vs. 46.51 in the control group, *p* < 0.001) and COX-2 (% mean change −37.76 in the intervention vs. 0.27 in the control group, *p* < 0.001) were reduced compared to the control group. sRAGE was reduced in both groups. In addition, HbA1c (at two months), total cholesterol, and triglycerides were reduced in the intervention versus the control group. The adoption of healthy cooking methods deserves further research as a possible way of modulating inflammatory markers in patients with CKD.

## 1. Introduction

Advanced glycation end products (AGEs) are formed in the presence of hyperglycaemia and diseases associated with high levels of oxidative stress, such as chronic kidney disease (CKD). In CKD, the higher levels of AGEs result from both increased formation and decreased renal clearance and may be related to survival [1]. AGEs contribute to the development and progression of CKD, and, conversely, the disease further promotes AGEs formation by inducing oxidative stress [2].

More specifically, AGEs play a key role in endothelial dysfunction, glomerulosclerosis, and renal function impairment, partly through interaction with their receptor (RAGE) [3,4]. On endothelial cells, RAGE inhibits the endothelial nitric oxide synthase and promotes endothelial dysfunction [5]. The AGE–RAGE interaction activates several signaling pathways [3]. Moreover, AGEs contribute to the activation of tumor necrosis factor-α (TNF-α), interleukin-1 (IL-1), insulin-like growth factor-1 (IGF-1), and platelet-derived growth factor (PDGF), which further promote collagen IV synthesis. They are also involved in the inflammatory response by increasing cyclooxygenase-2 (COX-2) levels, a key component of prostaglandin synthesis. COX-2 is expressed at low levels when renal function is normal and increases significantly as a response to inflammation and renal function impairment [6].

It is noteworthy that patients undergoing haemodialysis have significantly higher serum AGEs along with higher circulating soluble RAGE (sRAGE) levels [7]. Although the contribution of dietary glycotoxins to overall AGEs accumulation is well established, their role in CKD pathogenesis and progression has not been fully elucidated. The need for interventions, including improvement of cooking skills, is stated in the clinical practice dietary guidelines for nutrition in patients with CKD, but no recommendation is currently made on dietary AGEs intake [8]. The absence of such a recommendation may be due to the absence of high-quality evidence, particularly from intervention studies.

In this context, the clarification of the underlying mechanisms linking dietary glycotoxins to kidney pathophysiology is crucial for appropriately formulating and modifying CKD-related lifestyle and dietary recommendations. Thus, the aim of the present study was to investigate the effects of a diet restricted in AGEs on various biochemical and inflammatory indices in patients with end-stage renal disease (ESRD) undergoing dialysis.

## 2. Results

The basic characteristics of the intervention and control group are shown in Table 1. It is noted that the two groups did not differ in the body mass index (BMI) and treatment duration. However, the subjects in the control group were older. The results regarding several biochemical markers of the total studied population (study group and control group) are shown in Table 2 and Appendix A. The baseline values for most parameters did not differ between the intervention and control groups, except serum glucose and creatinine, which were higher in the intervention group. However, HbA1c did not differ between the groups. Regarding changes in the measured parameters at follow-up, several changes were observed in the intervention group (Table 2). In addition, HbA1c, total cholesterol, and triglycerides were reduced in the intervention group. It is of interest that analysis of covariance (ANCOVA), after adjusting for baseline triglycerides, had the same results (significant difference between the intervention and control groups). Notably, serum creatinine and calcium were increased in both groups, while the total protein was reduced in both groups. Moreover, fasting glucose was reduced in the control group (Table 2). When the mean or median differences of variables between the groups were compared, it was evident that HbA1c (at two months), total cholesterol, and triglycerides were significantly reduced in the intervention group compared to the control group (Table 2), Weight changes were not significant in the control and intervention group. Regarding selected measurements at two months follow-up, the differences were documented in the intervention group (reductions in HbA1c, total cholesterol, and triglycerides) whereas no change was recorded for the control group. In addition, the time measurements of C-reactive protein, HbA1c, and triglycerides differed between the control and intervention groups (repeated measurement analysis at baseline, one month follow-up, and two months follow-up, Appendix A).

The mean ± SD values or medians and interquartile ranges of the sRAGE, CML, RAGE, and COX-2 molecular markers are shown in Table 3. It is noteworthy that CML was higher in the intervention group at baseline (*p* < 0.001), whereas the other parameters did not differ. With regards to the effect of age on CML expression, the Spearman’s rho between age and baseline CML was 0.424 (*p* = 0.062) for the control group and −0.107 (*p* = 0.636) for the intervention group, but showed no significant correlation between age and CML. However, CML levels were compared between the control and the intervention groups, and across stratification by age, differentations were documented only for subjects ≤65 years (*p* = 0.002). 

Moreover, the authors stratified the sample according to age tertiles and compared the baseline CML of the control and the intervention groups. The number of patients in the aged group was higher in the control group (Chi square = 0.030).

At follow-up, sRAGE, RAGE, and COX-2 were reduced in the intervention group. Similarly, sRAGE was reduced at follow-up in the control group. CML was increased at follow-up in both the intervention and the control groups (Table 3). When the mean or median differences of variables between groups were compared, it was evident that the increase in CML was smaller in the intervention group. Significant decreases in RAGE and COX-2 were only observed in the intervention group, and these changes differed significantly from the control group (Table 3). 

An indicative image of the quantification of the protein expression of RAGE and COX-2 using densitometry (using Scion Image-version 4.0.3.2; Scion Corporation) is presented in Figure 1. At baseline, CML was negatively correlated with sRAGE (Spearman’s rho = −0.701, *p* < 0.001) and RAGE (Spearman’s rho = −0.346, *p* = 0.031) in the total sample. Moreover, RAGE was positively related to COX-2 (Spearman’s rho = 0.447, *p* = 0.004), cholesterol (Spearman’s rho = 0.376 * *p* = 0.018), and triglycerides (Spearman’s rho = 0.317, *p* = 0.049), while sRAGE was positively related to urea (Spearman’s rho = 0.311, *p* = 0.045). The correlations of follow-up values between CML, sRAGE, RAGE, and COX-2 and biochemical indices, in the control and intervention groups, are shown in Appendix A. Furthermore, the Spearman correlations were tested between the differences in sRAGE, CML, and RAGE with differences in glucose and lipid parameters in patients with and without diabetes, for the intervention and control group separately. The only significant correlation detected was that of differences in HbA1c at one month, with differences of sRAGE (Spearman’s rho = −0.900, *p* = 0.037), and CML (Spearman’s rho = 0.900, *p* = 0.037) in subjects with diabetes in the intervention group.

## 3. Discussion

This study presents the effects of a one-month controlled dietary intervention that restricted nutritional AGEs in patients with ESRD. The intervention group presented a lower increase in CML, compared to the control group, while RAGE and COX-2 were reduced compared to the control group. sRAGE was similarly reduced in both groups. In parallel, HbA1c (at two months), as well as total cholesterol and triglycerides, were significantly reduced in the intervention group compared to the control group.

According to our previous study, dietary AGEs play an important role in the progression of CKD as well as in the adverse prognosis of patients with ESRD [9]. Patients undergoing haemodialysis have significantly higher serum AGEs along with higher circulating sRAGE levels [7]. In patients with diabetes and renal disease, Busch et al. found that the accumulation of CML was not significant, whereas Wagner et al. showed an increase in overall CML levels [10,11].

Dietary AGEs can affect circulating AGEs in subjects with kidney disease [12]. In the present study, CML, a main representative of AGEs in tissues and plasma, was increased at follow-up in both groups. This implies that the sRAGE levels may not be sufficient to eliminate the excess levels of circulating CML. Other dietary interventions in patients with ESRD undergoing peritoneal dialysis showed that CML levels were lower after an AGEs-restricted diet compared to the pre-intervention levels, as opposed to the findings of the present study [12,13]. However, dietary AGEs alone do not significantly affect serum CML, since levels of serum or tissue AGEs, including CML, are influenced by age, glucose levels, protein metabolism, oxidative stress, inflammation, liver status, smoking, and renal function [14]. Indeed, a positive association of AGE peptides with serum creatinine has been documented [15,16]. This association is also present in patients with kidney failure [13]. In this context, it is possible that a deterioration in residual renal function in both groups (as shown by creatinine increase) may also have affected circulating AGEs, and CML. Notably though, the CML increase was lower in the intervention group compared to the control, reflecting the complex regulation of circulating AGEs.

In other patients, the data regarding CML concentrations and diet effects are also contradictory. An AGEs-restricted diet for two months did not affect total AGEs content in skin and urine in patients with diabetic nephropathy [17]. Other surveys have shown that dietary AGEs alone do not significantly affect serum CML, sRAGE and CRP [18,19], inflammatory markers or endothelial dysfunction [20].

Another factor that seems to influence the relationship between diets rich in AGEs and the circulating CML levels is the study duration. According to a systematic review of randomized controlled trials conducted by Clarke et al., the magnitude of the effect of a low-AGEs diet on circulating AGEs, such as CML, and inflammatory markers was different in short-term and long-term studies [21]. In addition, in some studies, urine AGEs were measured, reflecting the metabolic state of several hours preceding the sampling, while in the present study only circulating CML was measured, using an ELISA test [17].

The AGEs-restricted intervention in the present study reduced RAGE expression. The role of dietary AGES in RAGE expression has been also demonstrated in other studies. In a study by Vlassara et al., healthy controls and patients with CKD stage 3 followed either a low-AGEs diet or an unrestricted diet. A positive correlation was found between dietary AGEs and RAGE expression in both the patients with CKD and the controls [22].

One of the most interesting findings of the present study was the reduction in the sRAGE concentrations at follow-up in both studied groups, along with an increase in CML. It could be hypothesized that the sRAGE change may be a subsequent result of CML changes or a result of further impairment in residual renal function. However, this finding is not supported by previous literature data, according to which patients undergoing haemodialysis have higher serum AGEs and sRAGE levels [7], and CML concentrations that are positively correlated to sRAGE [23].

A study by Kalousová et al. has reported that serum sRAGE levels are increased in patients with impaired renal function, particularly in those with ESRD [24]. Jung et al. have also shown that sRAGE concentrations were elevated in patients with ESRD undergoing haemodialysis. Moreover, after kidney transplantation and restoration of the renal function, sRAGE concentrations were reduced [25]. Whether sRAGE is increased due to impaired residual renal function or due to its up-regulation to counteract the toxic effects of AGEs and other toxins accumulating in uremia remains to be clarified [24]. In addition, literature regarding the relationship between diet and sRAGE is limited. It could be assumed that since sRAGE has a protective role, diets with high concentrations of AGEs could elicit an increase in sRAGE plasma concentrations. However, an AGEs-restricted diet did not affect circulating sRAGE in patients with diabetic nephropathy [17].

The present study is the first to investigate the association of dietary AGEs with COX-2. COX-2 was reduced after the intervention with an AGEs-restricted diet, while RAGE expression was positively correlated to COX-2. To date, research has focused on COX-2 involvement in the development and progression of diabetes and its complications [26]. Increased glucose concentrations promote inflammation by regulating COX-2 expression through multiple signaling pathways, resulting in the activation of mononuclear cells, which are implicated in diabetes-related complications including CKD [26]. The same conclusion was reached by Giulietti et al., who found increased COX-2 concentrations in patients with diabetes mellitus [27]. The present intervention resulted in reductions in glucose levels, which may be related to reductions in COX-2 expression. AGEs may also modulate the inflammatory milieu by inducing changes in gut microbiota [28]. Interestingly, the inhibition of AGEs formation by glycation inhibitors reduces COX-2 in monocytes [29]. In addition, dietary AGEs increase COX-2 expression in mice, an effect counterbalanced by curcumin intake, underlying the possible beneficial effects of an antioxidant-rich diet [30]. Further studies on larger populations of patients with ESRD undergoing haemodialysis and of longer duration are needed to verify our findings.

In addition, patients who adopted a low-AGEs diet exhibited an improved glucose and lipid profile, which may have been (partly) responsible for the observed differences in AGE accumulation. Interestingly, metabolic benefits regarding HbA1c, cholesterol, and triglycerides persisted at the two months follow-up. Of note, RAGE protein expression was also correlated with total cholesterol and triglycerides. The relation between ingested glucose and circulating AGEs is relatively expected, considering the mechanistic aspects of AGEs formation [31]. Moreover, dietary AGEs restriction has been shown to improve insulin sensitivity in patients with type 2 diabetes [32]. As far as lipids are concerned, AGEs may be formed during lipid peroxidation [33] and a high-fat diet [18], although no association of lipidaemic indices with CML was documented in the present study. Possibly, an up-regulation of AGEs in the lipaemic state could lead to an increase in sRAGE [34]. Moreover, it is known that a diet that promotes hyperlipidaemia also promotes atherosclerosis and oxidative stress [35]. By binding to the RAGE receptor, AGEs activate NADPH oxidase leading to ROS production. The above is in line with our finding that a low-AGEs diet is associated with a lower sRAGE expression. A diet low in AGEs and rich in fresh products, such as the Mediterranean diet, involves the consumption of less ultra-processed foods [36] since food processing accelerates the formation of new AGEs [37]. Such a diet has beneficial effects on lipids, glucose regulation [38], and adiposity [36,38]. In addition, ultra-processed foods have been connected to a high prevalence of CKD [39,40], while inorganic phosphate additives may be related to the cardiorenal syndrome [41]. Nevertheless, certain additives may reduce formed AGEs in foods [42].

Regarding HbA1c, a decrease was observed in the intervention group at two and three months after the intervention. The connection between dietary AGEs and HbA1c levels is complex and involves multiple factors. Consuming a diet high in AGEs may contribute to increased glycation reactions. Indeed, dietary AGEs contribute significantly to the body’s AGEs pool, and individuals with high dietary intake of AGEs would also have higher serum HbA1c values [43,44]. Also, dietary AGEs are associated with inflammation and oxidative stress. Chronic inflammation and oxidative stress can damage pancreatic beta cells, and impair insulin sensitivity leading to elevated blood glucose levels and HbA1c over time [28]. Last but not least, the favorable changes in glycaemic and lipidaemic profile are not related to body weight changes, since weight remained relatively stable in both groups.

The limitations of the present study include the relatively small sample size. Moreover, medications such as aspirin may have modulated COX-2 levels and other indices. In addition, only serum CML was measured, while several other AGEs such as pentosidine or MG-H1, which is a methylglyoxal derivative, may be implicated [45]. Nonetheless, CML is one of the best- characterized AGE compounds, constituting a good AGE indicator [45]. Furthermore, CML was measured by the ELISA method. The gold-standard chemical method for CML measurement is mass spectrometry, which offers important advantages because of its high selectivity and precise detection of low CML concentrations. However, ELISA methods have been the most common approach for CML measurement in several matrices (serum or foods) both in clinical and basic research [46,47]. The difference in increase of CML is somewhat difficult to interpret since the control group had much lower baseline values, suggesting the possibility of “regression to the mean” phenomenon.

In the present study, high flux filters (Kuf > 20 mL/mmHg/h/1.0 m^2^) with synthetic membranes (1.9–2.2 m^2^) or low flux filters (Kuf ≤ 20 mL/mmHg/h/1.0 m^2^) with synthetic membranes (1.8–2.0 m^2^) were used. Filters may affect AGEs concentration. For example, after dialysis sessions, AGEs perturbations are mostly present in polysulfone filters but not in polymethacrylate filters [48]. The type of membranes did not change during the intervention and the percentage of patients who underwent haemodialysis with each type of membrane was the same in both interventions.

## 4. Materials and Methods

### 4.1. Study Design

This is a one-month dietary intervention in patients with ESRD undergoing haemodialysis. A convenience sample was used. Patients were recruited from the Haemodialysis Unit, General Hospital of Molaoi, Greece. Twenty-two participants constituted the intervention group and twenty participants constituted the control group. The selection criteria for the sample were as follows: (a) age > 18 years; (b) ability to communicate in Greek; (c) satisfactory level of understanding and cooperation; (d) absence of any active infection or neoplasia; and (e) willingness to participate in the study. Selected parameters were also measured at two months (uric acid, total cholesterol, triglycerides, erythrocyte sedimentation rate, and C-reactive protein) and three months (HbA1c) to investigate longer-term differences.

### 4.2. Ethical Approval

The research protocol was also approved by the Ethics Committee of the University of Peloponnese (6 December 2017). In addition, authorization was obtained by the ethical committee of the haemodialysis unit, in which the patients of the survey are monitored (protocol number: 7321/17-02-2016) and permission was obtained by the Hellenic Data Protection Authority (1759/ΓΝ/ΕΞ/6597-2/09-11/2016). The objectives of the study were explained in the first page of the questionnaire and written informed consent was obtained from the participants.

### 4.3. Dietary Intervention

For each participant in the intervention group, an individualized dietary plan was formulated with the aid of a nutritionist, which excluded foods high in AGEs according to a previous survey conducted by our group [9]. Alternative food choices with comparable nutritional value were proposed to replace high AGEs foods with low-AGEs foods. Moreover, emphasis was given to the avoidance of processing or cooking methods that enhance the formation of AGEs. More specifically, subjects were instructed to use the following cooking methods: boiling, poaching, stewing, or steaming. In addition, they were instructed to avoid frying and oven reheating. An example of the dietary recommendations given to the control group is provided in Appendix A. The caloric value and the nutritional value of the dietary plans were not altered. No dietary changes were applied to the control group. Throughout the intervention, the patients in the intervention arm recorded the type and quantities of foods and drinks consumed, as well as the cooking methods they used to prepare food, and were encouraged not to deviate from the dietary instructions. Patients’ questions were addressed through continuous personal contact or telephone communication with the researchers. The compliance was briefly checked through oral 24-h recalls with the patients before haemodialysis sessions.

### 4.4. Biochemical Measurements

For blood analysis, concerning haematological and biochemical markers, peripheral fasting blood samples were collected from all participants. Blood sampling was performed during the puncture of the arteriovenous access, before the start of the haemodialysis. Blood samples were obtained on the first day of the intervention and one month later. Studied markers were measured for all participants at both time points in the same hospital biochemical laboratory, immediately after blood draw. Haematological markers were measured in a tube with ethylenediamine tetraacetic acid (EDTA) using the Sysmex K-4500 (Toa Medical Electronics Co., Ltd., Kobe, Japan) and biochemical markers in blood serum using the analyzer Siemens Advia 1800 (Siemens Healthcare Diagnostics, Inc., Tokyo, Japan) through colorimetry and/or ion selective electrode (ISE).

Blood plasma was also isolated for further analysis. The plasma soluble form of the RAGE receptor (sRAGE) was measured using the sandwich ELISA (Human sRAGE, Soluble Receptor for Advanced Glycation End product, Fine-test, China (Sensitivity: < 18.75 pg/mL, Intra-Assay: CV < 8%, Inter-Assay: CV < 10%, Recovery range: 96%, cross reactivity: Specifically recognize sRAGE, no obvious cross reaction with other analogues). Carboxymethyl lysine (CML) was measured with a competitive ELISA Kit (Competitive-ELISA Human Carboxymethyl Lysine, Fine-test, China (Sensitivity: <9.375 ng/mL, Intra-Assay: CV < 8%, Inter-Assay: CV < 10%, Recovery range: 97% cross reactivity: Specifically, it recognizes CML with no obvious cross reaction with other analogues). According to the literature, for ELISA the inter-assay % CVs of less than 15 are generally acceptable and the intra-assay % CVs should be less than 10 [49,50]. For HbA1c, a third sample was obtained sixty (60) days after the first sampling to better capture potential changes. HbA1c in ethylenediamine tetraacetic acid (EDTA) plasma was analyzed by Ion-Exchange HPLC Technology (D-10 Hemoglobin Testing System, Biorad, CA, USA).

### 4.5. Western Immunoblotting

A second blood sample was obtained in a heparinized syringe (heparin/blood 90 IU/10 mL) and placed in a falcon tube with more than 5 mL of Ficoll–Paque solution. The blood sample was centrifuged (centrifuge conditions: 800× *g*, 4 °C, 10 min, acceleration: 9, brake: 0), resulting in the separation of four layers (top to bottom: plasma, mononuclear cells, Ficoll–Paque solution and erythrocytes). In the isolated mononuclear cells of the second layer, the protein expression of RAGE and COX-2 were studied using Western immunoblotting. Western blotting was performed as previously described [51]. The following dilutions and incubation times were applied: anti-RAGE (1:200, overnight at 4 °C) (Biotechnology, Santa Cruz, CA, USA), anti-Cox (1:1000, overnight, 4 °C) (Biotechnology, Santa Cruz, CA, USA) and anti-Tubulin (1:1000, room temperature), (Upstate New York, NY, USA). Quantification of the Western signals (complexed protein bands) was performed using the Bio-Rad Universal Hood II Gel Doc Imaging System (Bio-Rad Laboratories, Hercules, CA, USA) and the image analysis program Image-Pro Plus (version 4.5, Media Cybernetics, Rockville, MD, USA).

### 4.6. Statistical Analysis

Descriptive analysis of the haematological, biochemical, and molecular markers of the studied population was performed. The data are presented as means +/− standard deviation (for normal variables) or as medians and interquartile ranges (for non-normally distributed variables). Paired-t test or Wilcoxon sign test was used to compare the baseline and follow-up levels of normally distributed and non-normally distributed variables, correspondingly. In addition, the % differences of all variables were calculated for both groups. Then the mean or median differences (for normally or non-normally distributed differences, correspondingly) of the intervention and the control groups were compared with the t test or Mann–Whitney test. It is noted that several modifications were made to the non-parametric variables to meet the required conditions. Specifically, the following variables were logarithmized at baseline and follow-up: Erythrocyte sedimentation rate (ESR); creatinine; AST; CK; ALP; C; P; sRAGE. The following variables were reversed (1/variable) at baseline and follow-up: RBC; glucose; ALT; RDW; HbA1c; GGT. The following variables were squared at baseline and follow-up: MCV; MPV. For the case of triglycerides, analysis of covariance (ANCOVA) was also run, since large baseline differences were observed between the intervention and control group. In this case the % difference of triglycerides was set as a dependent variable, intervention was set as a fixed factor, and baseline triglycerides were set as covariate. Repeated-measures analysis of variance (RM-ANOVA) was used to compare changes of selected parameters measured at baseline, one month and two months, between the control and the intervention group. For C-reactive protein the Kruskal–Wallis test was performed to compare measurements, since the normality criterion was not fulfilled. Potential non-linear associations were assessed using Spearman’s rho correlations between the various clinical, biochemical, and molecular characteristics. The statistical analysis was performed using the statistical package SPSS v.24 (SPSS Inc., Chicago, IL, USA) and the significance level was set at 0.05.

## 5. Conclusions

In the clinical practice dietary guidelines for patients with CKD, it is underlined that there is a need for interventions regarding cooking skills [8]. Although these patients are already under many restrictions that affect their quality of life, the results of the present study can be used for the formulation of specific clinical practice advice toward more “kidney-friendly” cooking methods. Moreover, the present results underline the importance of the involvement of a multidisciplinary team in the care of patients with CKD.

In conclusion, the adoption of healthy cooking methods in the context of a low-AGEs diet deserves further research as a possible way of modulating lipid, glucose, and inflammatory indices in patients with ESRD. The accumulation of AGEs can be restricted directly (by adopting healthy eating habits) or indirectly (by modifying other risk factors), minimizing the AGEs-related complications such us systemic inflammation, oxidative stress, and progression of cardiovascular-disease-related mortality.

## Figures and Tables

**Figure 1 ijms-25-08893-f001:**
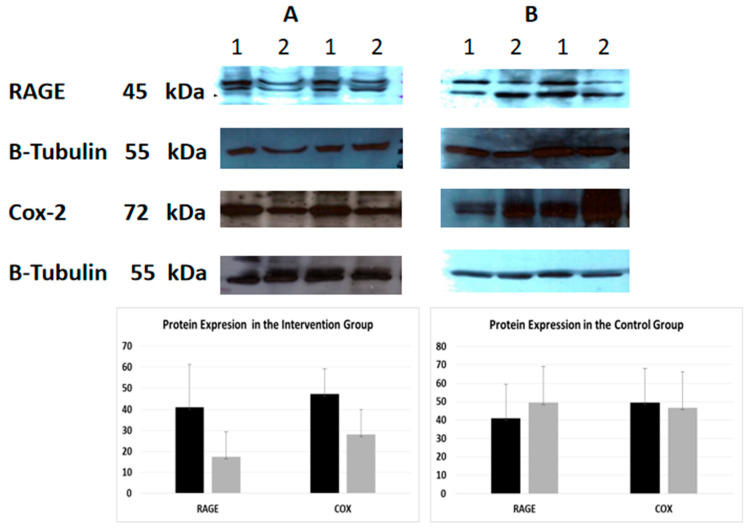
Protein expression of RAGE and COX-2 (Western Immunoblotting)—indicative images. Effect of intervention on RAGE and COX-2 protein expression in PMBC cells of patients. The protein expression level was measured by Western blot analysis at baseline, and one month after dietary intervention in equal amount of cell lysate (40 lg/lane) from isolated PMBC cells of each participant. Cell lysates was subjected to electrophoresis and analyzed by Western blot. Representative blots shown for each studied protein. (**A**). Intervention group (**B**). Control group. 1. Baseline 2. After intervention.

**Table 1 ijms-25-08893-t001:** Basic characteristics of participants.

	Intervention Group (*Ν* = 22) Mean ± Standard Deviation or Median (25th, 75th Percentile)	Control Group (*Ν* = 20) Mean ± Standard Deviation or Median (25th, 75th Percentile)	*p*-Value
	Baseline	Baseline	
Age (years)	66.9 ± 8.4	74.4 ± 9.2	0.009
BMI (kg/m^2^)	26.6 (24.3, 28.4)	24.4 (21.4, 28.0)	0.151
Patients with Diabetes (*n*)	6	6	1.000
Urea reduction rate (%)	62.2 ± 7.7	66.0 ± 8.28	0.132
Kt/V	1.21 ± 0.192	1.30 ± 0.19	0.134
Treatment duration (years)	4.0 (3.0, 6.0)	3.0 (2.0, 6.0)	0.476

Normally distributed variables are shown as mean± standard deviation. Non-normally distributed variables are shown as median and interquartile range.

**Table 2 ijms-25-08893-t002:** Haematological and Biochemical markers of the study population.

	Intervention Group (*Ν* = 22) Mean ± Standard Deviation or Median (25th, 75th Percentile)		*p*-Value	Control Group (*Ν* = 20) Mean ± Standard Deviation or Median (25th, 75th Percentile)		*p*-Value	% Difference * (Intervention)	% Difference * (Control)	*p*-Value
	Baseline	Follow-up		Baseline	Follow-up				
Markers of Renal Function									
Urea (mg/dL)	157 ± 39	166 ± 39	0.063	134 ± 46	135 ± 50	0.745	6.60 ± 14.61	1.56 ± 19.36	0.344
Creatinine (mg/dL) †	7.57 (5.83, 8.90) *	7.86 (6.20, 9.61)	0.001	5.92 (5.26, 6.85) *	6.12 (5.61, 7.20)	0.023	8.01 (3.13, 13.05)	9.59 (−1.43, 16.41)	0.632
Uric acid (mg/dL)	6.6 ± 1.3	6.8 ± 1.2	0.382	6.1 ± 1.1	6.1 ± 1.4	0.230	4.28 ± 15.51	0.57 ± 21.42	0.528
Glycemic profile									
Fasting glucose (mg/dL) †	107 (86, 131) *	103 (91, 149)	0.394	149 (103, 181) *	123 (87, 150)	0.018	3.23 (−11.52, 15.47)	−11.60 (−28.91, 2.89)	0.008
High fasting glucose (*n*, %)	5 (22.7%)	7 (31.8%)		13 (65.0%)	9 (45.0%)				
HbA1c (%) †	5.6 (5.1, 6.4)	5.4 (5.0, 6.1)	<0.001	5.7 (5.3, 7.1)	5.7 (5.3, 7.4)	0.770	−2.52 (−5.58, −0.98)	0.00 (−3.76, 4.85)	0.067
HbA1c (%) † (at 2 months)	5.6 (5.1, 6.4)	5.0 (4.7, 5.8)	<0.001	5.7 (5.3, 7.1)	5.8 (5.4, 7.5)	0.339	−8.44 (−11.32, −5.41)	1.90 (−2.97, 6.03)	<0.001
Inflammation Markers									
C-reactive protein, CRP (mg/dL)	0.3 (0.1, 0.7)	0.2 (0.1, 0.7)	0.224	0.5 (0.2, 1.0)	0.8 (0.3, 2.2)	0.247	−6.23 (−20.64, 27.24)	55.20 (−23.58, 102.57)	0.074
Erythrocyte sedimentation rate, ESR (mm/h) †	37 (24, 62)	30 (20, 50)	0.216	50 (41, 73)	55 (27, 62)	0.293	−34.26 (−16.66, 0.00)	−31.92 (−16.32, 22.14)	0.861
Lipid Profile									
Total cholesterol (mg/dL)	163 ± 42	149 ± 33	0.016	164 ± 39	164 ± 41	0.904	−7.17 ± 12.06	−0.05 ± 8.396	0.034
LDL-cholesterol (mg/dL)	70 ± 26	70 ± 20	0.373	79 ± 33	80 ± 36	0.601	−1.92 (−13.95, 15.22)	2.62 (−5.04, 14.50)	0.597
HDL-cholesterol (mg/dL)	48 ± 11	47 ± 10	0.853	47 ± 14	47 ± 15	0.819	−0.97 ± 11.05	0.81 ± 10.63	0.597
Triglycerides (mg/dL)	222 ± 88	167 ± 69	<0.001	188 ± 71	177 ± 71	0.252	−22.10 ± 20.60	−3.89 ± 26.49	0.017
Markers of nutritional status									
Total protein (g/dL)	6.9 ± 0.4	6.5 ± 0.4	<0.001	7.0 ± 0.3	6.5 ± 0.4	<0.001	−6.47 ± 3.62	−6.35 ± 4.67	0.705
Albumin (g/dL)	4.1 (3.9, 4.3)	4.1 (3.8, 4.3)	0.748	4.0 (3.8, 4.3)	4.0 (3.9, 4.2)	0.404	−1.13 (−4.87, 3.33)	−2.43 (−4.65, 3.22)	0.773
Dry weight (kg)	76.4 ± 12.7	76.5 ± 12.7	0.195	68.3 ± 17.9	68.1 ± 17.9	0.060	0.00 (0.00–0.27)	0.00 (−0.55–0.00)	0.028

Normally distributed variables are shown as mean ± standard deviation. Non-normally distributed variables are shown as median and interquartile range. † The variables were transformed as follows to achieve normality and conduct parametric tests for comparisons. Logarithmized variables (log variable) at baseline and follow-up: Erythrocyte sedimentation rate (ESR); creatinine. Reversed variables (1/variable) at baseline and follow-up: glucose; HbA1c. * Baseline value significantly different between intervention and control group.

**Table 3 ijms-25-08893-t003:** Molecular markers in the studied population.

	Intervention Group (*Ν* = 22) Mean ± Standard Deviation or Median (25th, 75th Percentile)		*p*-Value	Control Group (*Ν* = 20) Mean ± Standard Deviation or Median (25th, 75th Percentile)		*p*-Value	% Difference (Intervention)	% Difference (Control)	*p*-Value
	Baseline	Follow-up		Baseline	Follow-up				
sRAGE (pg/mL)	186 (89, 344)	103 (58, 189)	0.001	186 (140, 274)	82 (42, 124)	<0.001	−34.61 ± 32.84	−49.80 ± 24.45	0.100
CML (ng/mL)	284 (160, 353) *	347 (295, 371)	<0.001	93 (64, 132) *	279 (140, 377)	<0.001	12.39 (1.38, 56.89)	69.34 (23.40, 264.60)	0.013 **
RAGE protein expression	40.86 ± 20.27	17.47 ± 11.86	<0.001	40.82 ± 17.71	49.43 ± 18.69	0.074	−56.54 ± 20.07	46.51 ± 90.91	<0.001
COX-2 protein expression	47.27 ± 17.23	28.11 ± 11.56	<0.001	49.43 ± 18.69	46.70 ± 19.51	0.918	−37.76 ± 20.72	0.27 ± 14.653	<0.001

Mean value ± Standard Deviation. CML: carboxymethyl lysine; COX-2: cyclo-oxygenase 2; RAGE: Receptor for Advanced Glycation End products; sRAGE: Soluble Receptor for Advanced Glycation End products. * Baseline value significantly different between intervention and control group. ** Mean rank for % difference in control group 25.56; Mean rank for % difference in intervention group 16.36.

## Data Availability

Data are available upon request.

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
