# Peer review of "A One-Month Advanced Glycation End Products—Restricted Diet Improves CML, RAGE, Metabolic and Inflammatory Profile in Patients with End-Stage Renal Disease Undergoing Haemodialysis"

_ijms, 2024, doi:10.3390/ijms25168893_

Round 1

Reviewer 1 Report (New Reviewer)

Comments and Suggestions for Authors

In this paper the Authors studied the impact of 1-month controlled dietary nutritional restriction AGEs in patients with end-stage renal disease (ESRD) undergoing hemodialysis (n=22 participants in the intervention and n=20 participants in the control group). Hematological, biochemical markers, the soluble form of the receptor for AGEs (sRAGE), and carboxymethyl lysine (CML) were measured at baseline and follow-up end.

The intervention group presented a lower increase in carboxymethyl lysine (CML), a reduction of RAGE (% mean change -56.54 in the intervention vs 46.51 in the control group, p<0.001), COX-2, HbA1c (at 2 months),  total cholesterol and triglycerides, while sRAGE was reduced in both groups.

They concluded that the adoption of dietary habits including healthy cooking methods that reduce AGEs formation, results in an improvement of markers associated with cardiovascular complications.

General comment:

This is an excellent paper, providing new insights into the knowledge of AGEs role for hemodialysis patients, and what we likely have to do to improve the long-term prognosis of these patients. I have a few comments

Major comment

1) An alternative methodology of study would be a cross-over study. The Authors studied the patients at the one-month follow-up. However, I’m curious about the long-term outcome of the intervention group. Did the intervention group patients keep following the diet, or a part of they came back to the control diet? In such case, what was the outcome of clinical metabolic profile, such as HbA1c, total cholesterol, or triglycerides which significantly improved at one month?

2) Table 1 shows the baseline characteristics of the patients of intervention and control groups. However, dialysis efficiency is important in the studied data. Please add data such as the median Kt/V, urea reduction rate, the residual renal function (or urine output).

3) In Table 3, the value of RAGE protein expression of the Control group is 40.8 vs 49.3, with a % difference of 46±90. Are these data correct? Please check.

Comments on the Quality of English Language

good

Author Response

1 reviewer

In this paper the Authors studied the impact of 1-month controlled dietary nutritional restriction AGEs in patients with end-stage renal disease (ESRD) undergoing hemodialysis (n=22 participants in the intervention and n=20 participants in the control group). Hematological, biochemical markers, the soluble form of the receptor for AGEs (sRAGE), and carboxymethyl lysine (CML) were measured at baseline and follow-up end.

The intervention group presented a lower increase in carboxymethyl lysine (CML), a reduction of RAGE (% mean change -56.54 in the intervention vs 46.51 in the control group, p<0.001), COX-2, HbA1c (at 2 months),  total cholesterol and triglycerides, while sRAGE was reduced in both groups.

They concluded that the adoption of dietary habits including healthy cooking methods that reduce AGEs formation, results in an improvement of markers associated with cardiovascular complications.

 General comment:

This is an excellent paper, providing new insights into the knowledge of AGEs role for hemodialysis patients, and what we likely have to do to improve the long-term prognosis of these patients. I have a few comments

Major comment

  • An alternative methodology of study would be a cross-over study. The Authors studied the patients at the one-month follow-up. However, I’m curious about the long-term outcome of the intervention group. Did the intervention group patients keep following the diet, or a part of they came back to the control diet? In such case, what was the outcome of clinical metabolic profile, such as HbA1c, total cholesterol, or triglycerides which significantly improved at one month?

We would like to thank the reviewer for the comments. We agree with the reviewer that a cross-over design would be a good alternative for the present study, since each subject is the control of itself. We would like to clarify that the present study did not have a cross-over design but was a parallel study. Unfortunately, we do not have long term data on diet compliance and biochemical markers. Νevertheless HbA which was measured 2 months after the intervention was found to be reduced as mentioned in the text

  • Table 1 shows the baseline characteristics of the patients of intervention and control groups. However, dialysis efficiency is important in the studied data. Please add data such as the median Kt/V, urea reduction rate, the residual renal function (or urine output).

Thank you for this useful comment. We have added data regarding the urea reduction rate and the Kt/V in Table 1 Its is noted that no difference was detected between the control and intervention group regarding the two aforementioned parameters.

To calculate the residual renal function the urine output is required, which was not requested during the study.

3) In Table 3, the value of RAGE protein expression of the Control group is 40.8 vs 49.3, with a % difference of 46±90. Are these data correct? Please check.

Thank you for this comment. This data is correct. Two out of 20 control subjects had a big difference, which affects the mean percent difference (subject 1: bas RAGE 6,5 and follow up RAGE 26, subject 2 bas RAGE 14 and follow up RAGE 49, corresponding to a 300% and 251 % difference.

Reviewer 2 Report (New Reviewer)

Comments and Suggestions for Authors

Dear authors you will find my comments attached

Comments on the Quality of English Language

Author Response

2 Reviewer

This is a very interesting work. I studied it with attention and interest. Initially, I would suggest a change in structure, materials and methods must precede results.

Your results are described in detail and clearly. The discussion is extensive, to the point that you could present it as a narrative review, you can consider this idea. I would recommend noting the difficulty in these patients of dietary restrictions as these are patients already under many restrictions that affect the quality of their life. Approaching these patients is difficult and must be done carefully by multidisciplinary team in order to achieve their compliance I think such a report would be useful.

Thank you.

Thank you for your comments. Regarding the structure of the manuscript, we have followed the proposed template of the journal (mdpi.com/journal/ijms/instructions).

Regarding your second comment, we have now added in the manuscript that these patients are already under many restrictions affecting their quality of life and that a multidisciplinary team should be present in their care (Pls see Conclusions).

Reviewer 3 Report (New Reviewer)

Comments and Suggestions for Authors

Although the study lasted only 1 month, the present article is a valuable contribution in the field of the diet of chronic dialysis patients and deserves to be published. Indeed, although we recommend our patients to boil meat and vegetables and throw away the water and we also recommend them not to consume processed and canned foods for the motivation of K and P control, we forget to forbid them not to fry in oil or grill the raw materials of the diet.

I think that only a few small corrections are necessary (below)

Comments on the Quality of English Language

-          Line 254: instead of: ‘The relation of ingested glucose and circulating AGEs’ – I suggest : ‘The relation between ingested glucose and circulating AGEs’

-          Lines 263-264:  ‘our finding that a low-AGEs diet associated with a lower sRAGE expression’. Replace with ‘finding that a low-AGEs diet is associated with a lower sRAGE expression’

-          Line 339: instead of: ‘during the paracentesis of the arteriovenous access’ – I suggest: ‘during the puncture of the arteriovenous access’

-          Line 347: instead of: ‘Blood plasma was isolated as well for further analysis at the same time points sampling.’ – I suggest: ‘Blood plasma was isolated as well for further analysis at the same time points as sampling.’

-          Lines 348-349: instead of: ‘The plasma soluble form of the RAGE receptor (sRAGE) was measured in using the sandwich ELISA’ – I suggest: ‘The plasma soluble form of the RAGE receptor (sRAGE) was measured using the sandwich ELISA’

Author Response

3 Reviewer

Although the study lasted only 1 month, the present article is a valuable contribution in the field of the diet of chronic dialysis patients and deserves to be published. Indeed, although we recommend our patients to boil meat and vegetables and throw away the water and we also recommend them not to consume processed and canned foods for the motivation of K and P control, we forget to forbid them not to fry in oil or grill the raw materials of the diet.

I think that only a few small corrections are necessary (below)

We would like to thank the reviewer for the comments. Below we provide a point-by-point answer to the raised comments.

Comments on the Quality of English Language

-          Line 254: instead of: ‘The relation of ingested glucose and circulating AGEs’ – I suggest : ‘The relation between ingested glucose and circulating AGEs’

Done 

-          Lines 263-264:  ‘our finding that a low-AGEs diet associated with a lower sRAGE expression’. Replace with ‘finding that a low-AGEs diet is associated with a lower sRAGE expression’

Done

-          Line 339: instead of: ‘during the paracentesis of the arteriovenous access’ – I suggest: ‘during the puncture of the arteriovenous access’

Done

-          Line 347: instead of: ‘Blood plasma was isolated as well for further analysis at the same time points sampling.’ – I suggest: ‘Blood plasma was isolated as well for further analysis at the same time points as sampling.’

Done

 -          Lines 348-349: instead of: ‘The plasma soluble form of the RAGE receptor (sRAGE) was measured in using the sandwich ELISA’ – I suggest: ‘The plasma soluble form of the RAGE receptor (sRAGE) was measured using the sandwich ELISA’

Done

Reviewer 4 Report (New Reviewer)

Comments and Suggestions for Authors

See PDF in attach

Author Response

4 reviewer

We would like to thank the reviewer for the comments. Below we provide a point-by-point answer to the raised comments.

In this 4 week randomized, controlled study in a small group of Greek haemodialysis patients, the authors studied the influence of a diet low in nutritional AGEs on several indicators of AGE accumulation. This diet resulted in a decrease of cholesterol and HbA1c and a potentially beneficial decrease in RAGE and COX-2 expression in monocytes. There was also a lower increase of the only measured AGE carboxymethyl lysine (CML) in the intervention group. However, this difference in increase of CML is somewhat difficult to interpret because of the much lower baseline value in the control group and the possibility of regression to the mean as explanation for the observed difference in increase should be discussed in the results section.

The possibility of the “regression to the mean” effect has been added in the discussion (limitations).

The possibly detrimental effect of large and comparable decreases in sRAGE and the increases of CML in both groups was unexpected and the authors should try to explain this finding in the discussion.

Thank you for the comment we added to the discussion

One of the most interesting findings of our study was the reduction in the fol-low-up sRAGE concentrations in both studied groups along with CML increases. This change may be a subsequent result of CML changes or a result of impairment in renal function. Indeed, patients undergoing haemodialysis have both higher serum AGEs and sRAGE levels [7], while CML has been previously corelated to sRAGE[23]. In par-allel, deterioration in renal function may also be related to sRAGE increases. A study by Kalousová et al. has reported that serum sRAGE levels are increased in patients with impaired renal function particularly in those with ESRD [24]. Jung et al. have shown that sRAGE concentrations were elevated in patients with ESRD undergoing haemodialysis. Moreover, after kidney transplantation and restoration of the renal function of these patients, sRAGE concentrations were reduced

Overall, it is difficult to conclude that this diet has beneficial effect on AGE accumulation in haemodialysis patients and the conclusion in the last line of the abstract should be changed accordingly.

Thank you for this comment. The conclusion and abstract has been updated.

Other major comments:

− As indicated in the supplementary table 3, the instructions for the diet low in AGE were very drastic and complex and therefore probably difficult to comply to in daily life. The authors should present data on the way they supported the patients and checked their compliance.

Τhe authors had frequent meetings with patients since they were present in the hemodialysis sessions during the intervention. The compliance was briefly checked through oral 24-h recalls with the patients before every  hemodialysis sessions.

 In view of the drastic diet, one could expect a change in body weight and the authors should supply the body weights at baseline and follow-up in Table 2.

Thank you for the insightful comment. We have now included weight changes of the control and intervention group in Table 2. Weight did not change in either group (paired-t test). When comparing weight differences in the control and intervention group a significant difference was documented with patients of the intervention group having higher weight changes than those in the control group (increases in body weight). It is noted that the clinical significance of weight changes is low. In either case, the intervention group did not experience a reduction in body weight, which could metabolically favor the measured parameters. This point has been added in results and discussion.

In the discussion the influence of the diet, that was also low in fat, on glucose and lipid parameters deserves more attention because this effect could have been (partly) responsible for the observed differences in AGE accumulation.

 The fat content of the diet may play a role in circulating AGEs (Davis KE et al. Contribution of dietary advanced glycation end products (AGE) to circulating AGE: role of dietary fat. Br J Nutr. 2015 Dec 14;114(11):1797-806. doi: 10.1017/S0007114515003487). Especially, the heating of fat in the diet may multiply circulating AGEs (Uribarri J et al. Advanced glycation end products in foods and a practical guide to their reduction in the diet. J Am Diet Assoc. 2010 Jun;110(6):911-16.e12. doi: 10.1016/j.jada.2010.03.018).

The present intervention did not intend to reduce total fat content of the diet, so the intervention was not necessarily low in fat. Patients were instructed to consume low fat dairy and to avoid fried foods. However, the use of olive oil in salads was encouraged.

We also agree with the reviewer that the induced changes in lipid and glucose (caused by the intervention) may be responsible for the observed changes in AGEs. This has been added in the discussion.

− Since age influences serum and tissue AGEs including CML, the authors should test if the difference in age between the 2 groups was responsible for the large difference in baseline CML.

The spearman rho between age and baseline CML was 0.424 (p=0.062) for the control group and -0.107 (p=0.636) for the intervention group. In this way, no significant correlation is documented between age and CML in the present sample.

Moreover, the authors stratified the sample according to age tertiles and compared baseline CML of the control and intervention group. The number of patients in the aged group was higher in the control group (Chi square= 0.030).

Intervention * age (Binned) Crosstabulation

Count  

age (Binned)

Total

<= 65.00

66.00 - 76.00

77.00+

Intervention

.00

5

6

9

20

1.00

10

10

2

22

Total

15

16

11

42

However, when CML levels were compared between the control and intervention group across age strata differentiations were documented only for subjects <=65 years (p=0.002). Descriptive data across strata are shown below.

Statisticsa

CML_bas_NP  

N

Valid

5

Missing

0

Percentiles

25

58.5350

50

81.5200

75

84.4650

a. age (Binned) = <= 65.00, Intervention = .00

Statisticsa

CML_bas_NP  

N

Valid

10

Missing

0

Percentiles

25

162.7850

50

348.9950

75

358.6775

a. age (Binned) = <= 65.00, Intervention = 1.00

Statisticsa

CML_bas_NP  

N

Valid

6

Missing

0

Percentiles

25

47.5450

50

107.3550

75

353.9500

a. age (Binned) = 66.00 - 76.00, Intervention = .00

Statisticsa

CML_bas_NP  

N

Valid

10

Missing

0

Percentiles

25

153.3075

50

274.0800

75

337.2700

a. age (Binned) = 66.00 - 76.00, Intervention = 1.00

Statisticsa

CML_bas_NP  

N

Valid

9

Missing

0

Percentiles

25

83.1650

50

111.3100

75

178.8600

a. age (Binned) = 77.00+, Intervention = .00

Statisticsa

CML_bas_NP  

N

Valid

2

Missing

0

Percentiles

25

136.2700

50

208.3800

75

.

a. age (Binned) = 77.00+, Intervention = 1.00

− Since approximately 25% of the patients in both groups were diabetics, the authors should perform an exploratory analysis to detect a possible difference in effect of the AGE restricted diet on both glucose and lipid parameters as well as AGEs

The authors have performed additional correlations regarding differences in AGEs and glucose, lipid parameters. More particularly, the Spearman correlations were tested between the differences in sRAGE, CML and RAGE with differences in glucose and lipid parameters in patients with and without diabetes, for the intervention and control group separately. The only significant correlation detected was that of differences in HbA1c at one month with differences of sRAGE (Spearman rho= -0.900, p=.037) and CML (Spearman rho=0.900, p=0.037) in subjects with diabetes in the intervention group. This results were added to the results.

− Since the type of artificial kidney could influence AGE production, the authors should supply data on the type of membrane in the 2 groups and possible changes during the study

In the present study high flux filters (Kuf>20ml/ mmHg/ h/ 1.0 m2) with synthetic membranes (1.9-2.2 m2) or low flux filters (Kuf≤20ml/ mmHg/ h/ 1.0 m2) with synthetic membranes (1.8-2.0 m2) were used.

The type of membranes did not change during the intervention and the percentage of patients who underwent hemodialysis with each type of membrane was the same in both interventions

− Both introduction and discussion should be shortened and concentrate on the subject of this

study (being effect of the diet in haemodialysis patients) and the results in this patient group

The introduction and discussion have been shortened according to the reviewer’s comment.

− Table 2 in the main text should only contain the laboratory values that could be expected to be changed by the diet being renal function, glycaemic profile, lipid profile and markers of nutritional status including body weight. For non-parametric variables, only median and 25 th and 75th percentile would suffice (min and max could be skipped). Other laboratory values should be supplied in a table in supplementary material

Table 2 has been accordingly updated. Several data on hematologic characteristics have been transferred to Supplementary files.

− Table 4 supplies correlations between baseline variables in the whole group and are not really relevant for the subject of this study and should therefore be omitted. The only relevant correlations are the very high negative correlations of approximately -0.96 between CML and sRAGE at follow-up in both groups, that are supplied in supplementary material.

Table 4 has been omitted and some data have been kept in the text.

Minor comments:

− Line 98-99: text should be changed to “significant decreases in RAGE and COX-2 were only observed in the intervention group and these changes differed significantly from the control group”

The authors have made the proposed change.

− Table 2 median value of calcium level at follow-up in the intervention group is probably wrong because 9.2 is not between the 25th and 75th percentiles of 9.6 and 9.9

Thank you for this observation. The value for calcium has been corrected (supplementary file).

− Lines 171-173: it is unlikely that the in haemodialysis patients (reflecting a small decrease in residual renal function) are responsible for the large increases in CML.

We would like to thank the reviewer for this comment. Of course, we cannot be sure about the reasons why CML increased in the present study. The co­efficient of variation of serum creatinine levels is relatively small; therefore, changes of 0.3 mg/dl or more are most probably not a result of measurement error and have been previously proposed as a diagnostic criteria of stage 1 acute kidney injury (Molitoris et al, nature clinical practice NEPHROLOGY, 2007 and KDIGO 2022).

 Moreover, several lines of evidence support the relation of creatinine and CML.  Results from the Baltimore study also show that persons with CKD and creatinine 1.30 (1.10, 1.50) had higher serum CML than persons without CKD and creatinine 0.90 (0.80, 1.10) implying that even small differences in creatinine are related to differences in CML (Semba RD et al. Serum carboxymethyl-lysine, a dominant advanced glycation end product, is associated with chronic kidney disease: the Baltimore longitudinal study of aging. J Ren Nutr. 2010 Mar;20(2):74-81. doi: 10.1053/j.jrn.2009.08.001.).

In addition, previous studies have shown a good correlation between circulating AGEs and serum creatinine (standardized β in univariate analysis = 0.608; P < 0.0001) suggesting that one unit change in circulating AGEs relates to a 0.6 mg/dl change in serum creatinine  (Schalkwijk CG et al. Plasma levels of AGE peptides in type 1 diabetic patients are associated with serum creatinine and not with albumin excretion rate: possible role of AGE peptide-associated endothelial dysfunction. Ann N Y Acad Sci. 2005 Jun;1043:662-70).

− Lines 175-183: the high fat content in the mediterranean diet cannot explain the increase in CML

in the control group wherein the diet wasn’t changed and this paragraph should therefort be removed

The requested change was made.

− Lines 202-214: these data are not relevant for this study in haemodialysis patients

These data were deleted according to the reviewer’s suggestion.

− Lines 215-236: discussion should be focused on the observed decreases of sRAGE and increases of CML in both groups of haemodialysis patients and not on progression of renal disease and other effects of sRAGE

The text has been reformulated in order to reflect all possible reasons why sRAGE increases may be present.

− Lines 237-251: in this discussion, the possibility that the effect of the AGE-restricted diet on COX2 expression was a consequence of the influence of the diet on glucose metabolism (as discussed in the next paragraph) and therefore an indirect effect

Thank you for this comment. We have more clearly stated the idea that the intervention reduced circulating glucose, and this could possibly reduce COX-2 expression.

− Lines 264-270: the influence of a Mediterranean diet on several parameters is not relevant since both groups used this diet before and during the study

The authors would like to underline that the intervention group was advised to avoid fried foods and ultra-processed foods, including pizza etc, which inevitably is a healthier dietary pattern, possibly closer to the mediterranean diet. The control group received no dietetic advice, while the exact degree of Mediterranean diet adherence of both groups is not known. The authors mostly underline that the consumption of ultra-processed foods (related to adverse health effects) was lower in the intervention group.

− Lines 279-282: this explanation is not applicable to this patients on haemodialysis

The authors have deleted the text in lines 279-282.

− Lines 283-287: not relevant for this study

The authors have deleted the text in lines 283-287.

Round 2

Reviewer 1 Report (New Reviewer)

Comments and Suggestions for Authors

For the completeness of the work, it would have been really useful to have two-month follow-up data, even just on common clinical parameters such as those asked for (triglycerides, cholesterol, etc.), and which are used in routine follow-up. Since the Authors cannot provide these data, and there is no demonstration of these improvements over time, I correctly suggest changing the title of the work to: A one-month advanced glycation end products -restricted diet improves CML, RAGE, metabolic and inflammatory profile in patients with end-stage renal disease undergoing haemodialysis"

Comments on the Quality of English Language

good

Author Response

For the completeness of the work, it would have been really useful to have two-month follow-up data, even just on common clinical parameters such as those asked for (triglycerides, cholesterol, etc.), and which are used in routine follow-up. Since the Authors cannot provide these data, and there is no demonstration of these improvements over time, I correctly suggest changing the title of the work to: A one-month advanced glycation end products -restricted diet improves CML, RAGE, metabolic and inflammatory profile in patients with end-stage renal disease undergoing haemodialysis”

We would like to thank the reviewer for the insightful comments. We have now retrospectively collected selected data at 2 months (for uric acid, total-cholesterol, triglycerides, erythrocyte sedimentation rate, and C-reactive protein and 3 months (for HbA1c). Interestingly, favorable changes in the lipid profile persisted over 2 months, changes in HbA1c persisted over 3 months and C-reactive protein was reduced in the intervention group compared to the control group (please see the attached table for more details). We have incorporated some results in the manuscript (purple text) and have attached the table as a supplementary one.

Moreover, we have changed the title as suggested.

Reviewer 4 Report (New Reviewer)

Comments and Suggestions for Authors

The authors have supplied satisfying answers to almost all my comments and have changed the draft manuscript according to my suggestions.

There are only a few minor issues that have to be implemented to make this manuscript acceptable for publication:

- in haemodialysis patients the serum creatinine reflects both muscle mass as well as residual renal function. Therefore, in the text "residual" should be added to "renal function" on lines 192, 218 (further impairment in residual renal function), 220, and 226

- lines 219-225: since reduction in (residual) renal function in the literature is related to sRAGE increases, a further reduction in renal function in both groups in this study cannot explain the observed decreases in sRAGE. The authors should change the text accordingly

- Table 1 and line 73: treatment duration should not be removed, since the result is important and mentioned in the text

- Line 113-114: the authors should add the information that there was no correlation between the difference in age and CML between the 2 groups (as mentioned with an additional analysis in their response letter)

Author Response

Dear reviewer

Thank you for your comments and the opportunity to improve our paper, all the changes are depicted in yellow and you can follow them  through track changes.

The authors have supplied satisfying answers to almost all my comments and have changed the draft manuscript according to my suggestions.

There are only a few minor issues that have to be implemented to make this manuscript acceptable for publication:

- in haemodialysis patients the serum creatinine reflects both muscle mass as well as residual renal function. Therefore, in the text "residual" should be added to "renal function" on lines 192, 218 (further impairment in residual renal function), 220, and 226

Done

- lines 219-225: since reduction in (residual) renal function in the literature is related to sRAGE increases, a further reduction in renal function in both groups in this study cannot explain the observed decreases in sRAGE. The authors should change the text accordingly

The phrase was replaced

- Table 1 and line 73: treatment duration should not be removed, since the result is important and mentioned in the text

The line in the table 1 was restored

- Line 113-114: the authors should add the information that there was no correlation between the difference in age and CML between the 2 groups (as mentioned with an additional analysis in their response letter)

Data were added to the text of the paper

Sincerely yours

Andrea Paola Rojas

Round 3

Reviewer 1 Report (New Reviewer)

Comments and Suggestions for Authors

I have no further comments

Comments on the Quality of English Language

good

Author Response

Dear reviewer

Thank you for your comments and the opportunity to improve our paper. English editing was  done.  

All the changes are depicted in yellow and you can follow them also through track changes.

Sincerely yours

Andrea Paola Rojas

This manuscript is a resubmission of an earlier submission. The following is a list of the peer review reports and author responses from that submission.

Round 1

Reviewer 1 Report

Comments and Suggestions for Authors

This is a fine work, nevertheless, some questions should be addressed before to be accepted. The firs of all, how many diabetic patients were included in each group? 

Discussion is confusing and it needs to be re-writen with better order. First at all, CML en AGE products are used as synonimous and CML is just one of them selected in this research. So that, the results should be discussed separately. In would be interesting to talk about the relationship of carbamilated end products and AGE products, who where measured together in earlier studies. 

Statistics: Since baseline value of triglyceride were far higher in the intervention group, paired t test is not adequate, ANCOVA would be needed to confirm the diferences.

"It is possible that both the control and inter- vention groups in the present study included a large amount of fat in the diet since both groups live in a Mediterranean country and the Mediterranean diet is recommended for renal patients]." Please, explain this. Mediterranean diet is a low fat diet 155 156 157

Conclusions are not related to the results, particularly "

Indeed, a euglycemic state, prevention of obesity and   
improved endothelial function may constitute alternative ways of minimizing the endog-  
enous formation of AGEs. Such approaches are particularly important in patients with 331
ESRD undergoing hemodialysis due to reduced excretion of AGEs through the urinary 332
system.

Comments on the Quality of English Language

n/a

Reviewer 2 Report

Comments and Suggestions for Authors

The authors present a study which tries to convince the reader that diet intervention may help to reduce AGEs

1. The assays used for measuring AGEs are questionable. One would expect a mass spectrometry based test in such a study, but we see how a cheap obscure Chinese ELISA has been used. This ELISA has not neither been validated nor referenced. Cross-reactivity, LOQ, LOD, within- and between-run CV : nothing is known!

2. Methodology is not the authors' strong point. HbA1c methodology is lacking.

3.The analyte description is EXTREMELY SLOPPY:

4. creatine kinase is still designated as "phosphokinase" a term which has been abolished in 1970. Surprisingly, not an activity is measured but a mass concentration (mg/dl, which is impossible since CK represents an iso enzyme system). Where IU/L roughly corresponds to 1 µg/L, we read here concentrations which are about 1,000,000 (1 million) times higher than normal (this means a lethal rhabdomyolyis). Surprisingly a 4 digit precision is depicted.

5. For AST and ALT the old abbreviations SGOT and SGPT have been used (abolished 50 years ago) with an impossible precison as well

6.The subjects enrolled in the study must be terribly ill since bilirubin concentrations in g/dl are given. Innormal subjects, this is mg/dl (thousand tiles less). These reported values represent a lethal icterus as well.

7.The patients are unlikely to be humans since also their calcium concentrations are lethal. Normal calcium levels are 10 mg/dl or 2.5 mg/dl, but here lethal values of 10 mmol/L are reported (not compatible with life!)

8. In agreement with the previous points, the authors do no care about assay precision as well: values for ESR, chol, HDL, trig, CML, alkaline phosphatase are all given with an impossible 4 digit precision, sodium even with a 5 digit precision.

9. Table 3: in case of non -Gaussian distributions, SD has no physical meaning: e.g. 17 ± 11 should be given as median and IQR and not as mean and SD

10. No information about the pre-analytical phase is given (e.g fasting, specimen transport, sample handling,....)

11. The biological variation of CML seems to be very high (not matched between the groups points towards a systematic error)

12. The difference between the presented data and ref 16 (a non Greek population) is ascribed to the Greek diet. This is a simple explanation but does not take into account the methodological differences (urine has a memory effect and reflects the metabolic state of several hours preceding the sampling, the ELISA test only gives the last minutes preceding the sampling as CML is cleared rapidly by the kidney and is characterized by a huge biological variation as can be read in table 3 (non-matching is likely due to biological variation and not intervention).

13. The molecular mechanism how HbA1c can be affected is mysterious: In HbA1c the N terminal valine  reacts with glucose: how can exogenous AGEs be involved in this very selective reaction? This is very remarkable since fasting plasma glucose values hardly changed. The authors should provide a molecular mechanism for this as this appears to violate the law of chemical mass action!

Comments on the Quality of English Language

no comments

Round 2

Reviewer 1 Report

Comments and Suggestions for Authors

All queries have been resolved.

Comments on the Quality of English Language

NA

Author Response

We thank the reviewer for the revision. We have made several corrections regarding the language.  New corrections are depicted in blue color

Reviewer 2 Report

Comments and Suggestions for Authors

The errors regarding the measuring units have been corrected

still impossible 5 significant digits are reported for CML, which has been measured by an obscure , unvalidated Chinese ELISA (with a CV of 8 to 10%, as can be read in the materials and methods, proving that the authors do not have  a critical attitude vs their own methods). In a matrix of CKD patients, ELISA have the greatest problems, but this is ignored

the official abbreviation of creatine kinase is CK (not CPK, which has been abolished in 1970)

Author Response

Comment: The errors regarding the measuring units have been corrected, still impossible 5 significant digits are reported for CML which has been measured by an obscure , unvalidated Chinese ELISA (with a CV of 8 to 10%, as can be read in the materials and methods, proving that the authors do not have  a critical attitude vs their own methods).

Answer:

The digits of CML and other variables have been corrected.

According to the literature, Inter-assay % CVs of less than 15 are generally acceptable and Intra-assay % CVs should be less than 10.

Brindle E, Lillis L, Barney R, Bansil P, Hess SY, Wessells KR, Ouédraogo CT, Arredondo F, Barker MK, Craft NE, Fischer C, Graham JL, Havel PJ, Karakochuk CD, Zhang M, Mussai EX, Mapango C, Randolph JM, Wander K, Pfeiffer CM, Murphy E, Boyle DS. A multicenter analytical performance evaluation of a multiplexed immunoarray for the simultaneous measurement of biomarkers of micronutrient deficiency, inflammation and malarial antigenemia. PLoS One. 2021 Nov 4;16(11):e0259509. doi: 10.1371/journal.pone.0259509.

Comment: In a matrix of CKD patients, ELISA have the greatest problems, but this is ignored

The gold-standard chemical method for CML measurement is mass spectrometry, which offers important advantages because of its high selectivity and precise detection of low CML concentrations. However, ELISA methods have been the most common approach for CML measurement in serum or foods, both in clinical and basic research. This comment was added to the limitation of the study

( Gómez-Ojeda A, Jaramillo-Ortíz S, Wrobel K, Wrobel K, Barbosa-Sabanero G, Luevano-Contreras C, de la Maza MP, Uribarri J, Del Castillo MD, Garay-Sevilla ME. Comparative evaluation of three different ELISA assays and HPLC-ESI-ITMS/MS for the analysis of Nε-carboxymethyl lysine in food samples. Food Chem. 2018 Mar 15;243:11-18. doi: 10.1016/j.foodchem.2017.09.098./ Detection of Noncarboxymethyllysine and Carboxymethyllysine Advanced Glycation End Products (AGE) in Serum of Diabetic Patientshttps://molmed.biomedcentral.com/articles/10.1007/BF03402128

Comment: the official abbreviation of creatine kinase is CK (not CPK, which has been abolished in 1970)

Answer: As shown below, CPK is still the abbreviation provided by the laboratory used for the measurement of creatine kinase. This is also true for many laboratories in Greece. However, we agree with the reviewer’s comment and have changed “CPK” to “CK”.

New corrections are depicted in blue color
